# GAMA: Generative Adversarial Multi-Object Scene Attacks

**Abhishek Aich**[*] **Calvin-Khang Ta**[*]**, Akash Gupta, Chengyu Song,**
**Srikanth V. Krishnamurthy, M. Salman Asif, Amit K. Roy-Chowdhury**
University of California, Riverside, CA, USA

## Abstract

The majority of methods for crafting adversarial attacks have focused on scenes with a single dominant object (e.g., images from ImageNet). On the other hand, natural scenes include multiple dominant objects that are semantically related. Thus, it is crucial to explore designing attack strategies that look beyond learning on single-object scenes or attack single-object victim classifiers. Due to their inherent property of strong transferability of perturbations to unknown models, this paper presents the first approach of using generative models for adversarial attacks on multi-object scenes. In order to represent the relationships between different objects in the input scene, we leverage upon the open-sourced pre-trained vision-language model CLIP (Contrastive Language-Image Pre-training), with the motivation to exploit the encoded semantics in the language space along with the visual space. We call this attack approach Generative Adversarial Multi-object Attacks (**GAMA**). **GAMA** demonstrates the utility of the CLIP model as an attacker's tool to train formidable perturbation generators for multi-object scenes. Using the joint image-text features to train the generator, we show that **GAMA** can craft potent transferable perturbations in order to fool victim classifiers in various attack settings. For example, **GAMA** triggers ∼16% more misclassification than state-of-the-art generative approaches in black-box settings where both the classifier architecture and data distribution of the attacker are different from the victim. Our code is available here: https://abhishekaich27.github.io/gama.html

## 1 Introduction

Despite attaining significant results, decision-making of deep neural network models is brittle and can be surprisingly manipulated with adversarial attacks that add highly imperceptible perturbations to the system inputs [1, 2]. This has led to dedicated research in designing diverse types of adversarial attacks that lead to highly incorrect decisions on diverse state-of-the-art classifiers [2–13]. The majority of such adversarial attacks [2, 8–18] has focused on scenes with a single dominant object (*e.g.*, images from ImageNet [19]). However, natural scenes consist of multiple dominant objects that are semantically associated [20–25]. This calls for attack methods that are effective in such multi-object scenes.

A recent body of work in adversarial attacks [26–30] has shown the importance of exploring attack methodologies for real-world scenes (although designed for attacking object detectors). However, such methods are image-specific approaches that are known to have poor time complexity when perturbing large batches of images, as well as poor transferability to unknown models (more details in Section 2) due to their inherent property of perturbing images independently from one another.

---

[*]Equal contribution. Corresponding author: AA (aaich001@ucr.edu). AG is currently with Vimaan AI, USA.

Different from such approaches, *our interest lies in the generative model-based approaches* [10–13] which are distribution-driven and craft perturbations by learning to fool a surrogate classifier for a large number of images. These generative adversarial attacks show stronger transferability of perturbations to unknown victim models and can perturb large batches of images in one forward pass through the generator demonstrating better time complexity [11, 31]. However, these generative attack methods have focused on learning from single-object scenes (*e.g.*, ImageNet in [11–13], CUB-200-2011 [32] in [13]) or against single-object surrogate classifiers (*e.g.*, ImageNet classifiers [33] in [10–13]). When trained against multi-object (also known as multi-label) classifiers to learn perturbations on multi-object scenes, such methods perform poorly as they do not explicitly incorporate object semantics in the generator training (see Table 2 and Table 3). As real-world scenes usually consist of multi-object images, designing such attacks is of importance to victim model users that analyze complex scenes for making reliable decisions *e.g.* self-driving cars [34]. To this end, *we propose the first generative attack approach, called* **G**enerative **A**dversarial **M**ulti-object scene **A**ttacks or **GAMA**, *that focuses on adversarial attacks on multi-object scenes.*

Progress in recent vision-and-language (VL) models [35–39] that allow joint modelling of image and text have garnered interest in recent times due to their versatile applicability in various image downstream tasks like inpainting, editing, *etc.* [40–51]. For the first time in literature, we introduce the utility of a pre-trained open-source framework of the popular VL model named CLIP (Contrastive Language-Image Pre-training) [36] in generating adversarial attacks. Trained on 400 million image-text pairs collected from the internet, CLIP has been shown to provide robust joint representations of VL semantics [40, 46] and strong zero-shot image classification on diverse datasets [36, 44]. This allows us to access diverse

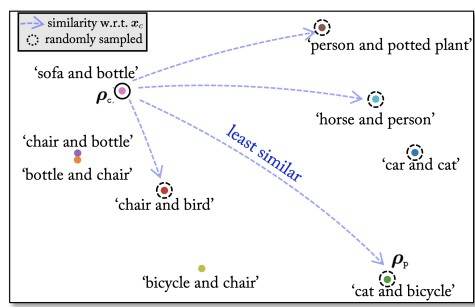

Figure 1: Using CLIP's image-text aligning property, we compute the features of the least similar text description w.r.t. to clean image.

VL features cheaply without any training as end-user. Our proposed **GAMA** attack employs the CLIP model to exploit the natural language semantics encoded in text features along with the vision features (due to its joint image-text alignment property). Different from prior works, **GAMA** utilizes CLIP model's extracted knowledge from ∼400 million images to maximize the feature differences of perturbed image $x_p$ against two different types of features computed from clean image $x_c$: (*1*) features of $x_c$ computed from surrogate models, and (*2*) features of $x_c$ computed from CLIP's image encoder. Additionally, **GAMA** also guides $x_p$ to contain different features compared to $x_c$ by using features from CLIP's text encoder *via* a contrastive loss function. For example in Figure 1, consider a clean image $x_c$ with objects "sofa and bottle". Using CLIP's image-text aligning property, we estimate that $x_c$ (with text features $\rho_c$) is least similar to the text prompt "car and bicycle" (text features $\rho_p$) among some randomly chosen candidates (indicated by dotted circles). **GAMA** uses $\rho_p$, created from a contextually consistent classes, to contrast and move the perturbed $x_p$ away from $x_c$ in feature space. Hence, the perturbed image features are comparably robust to data distribution changes in victim models as $\mathcal{G}_\theta(\cdot)$ is optimized to create perturbations that differ in features from two different image features. This allows **GAMA** to launch highly transferable attacks on unseen victim models (see Section 4). To summarize, we make the following contributions in this paper.

1. **Multi-object scene based generative attack aided by VL models.** We propose the first multi-object scene based generative attack, **GAMA**, that is designed to consider object semantics through vision-and-language models.

2. **Pre-trained CLIP model as an attacker's tool.** We propose the first generative attack on classifiers that utilizes the open-source pre-trained CLIP model as an attacker's tool to train perturbation generators.

3. **Extensive Attack Evaluations.** Our extensive experiments on various black-box settings (where victims are multi-label/single-label classifiers and object detectors) show **GAMA**'s state-of-the-art transferability of perturbations (Table 2, 3, 5, 4, 6, and 7). Additionally, we also show that **GAMA** outperforms its baselines in terms of attack robustness when the victim deploys state-of-the-art defenses (Table 8).

Table 1: **Characteristic comparison.** Here, $\boldsymbol{f}(\cdot)$ denotes the surrogate classifier. $\boldsymbol{x}$ and $\boldsymbol{x_\delta}$ denote a clean and perturbed image. $k$ denotes output from a specific pre-defined layer of $\boldsymbol{f}(\cdot)$ (different for each method). Better than prior generative attacks [10–13], **GAMA** leverages multi-modal (text and image) features $\boldsymbol{\rho}_{\texttt{txt}}$ and $\boldsymbol{\rho}_{\texttt{img}}$ extracted from a pre-trained CLIP [36] model for train the perturbation generator. Its learning objective aims to pull $\boldsymbol{f}_k(\boldsymbol{x_\delta})$ closer to a dissimilar text embedding $\boldsymbol{\rho}_{\texttt{txt}}$ (w.r.t. $\boldsymbol{x}$) while pushing it away from $\boldsymbol{f}_k(\boldsymbol{x})$ and $\boldsymbol{\rho}_{\texttt{img}}$. Further, **GAMA** analyzes attack scenarios where the surrogate model is a multi-label classifier with input scenes that usually contain multiple objects.

| Attack | Venue | Generator training strategy | Analyzed input scene? |
|:---:|:---:|:---:|:---:|
| GAP [10] | CVPR2018 | maximize difference of $\boldsymbol{f}(\boldsymbol{x_\delta})$ and $\boldsymbol{f}(\boldsymbol{x})$ | single object |
| CDA [11] | NeurIPS2019 | maximize difference of $\boldsymbol{f}(\boldsymbol{x_\delta})$ - $\boldsymbol{f}(\boldsymbol{x})$ and $\boldsymbol{f}(\boldsymbol{x})$ | single object |
| TAP [12] | NeurIPS2021 | maximize difference of $\boldsymbol{f}_k(\boldsymbol{x_\delta})$ and $\boldsymbol{f}_k(\boldsymbol{x})$ | single object |
| BIA [13] | ICLR2022 | maximize difference of $\boldsymbol{f}_k(\boldsymbol{x_\delta})$ and $\boldsymbol{f}_k(\boldsymbol{x})$ | single object |
| **GAMA** | Ours | contrast $\boldsymbol{f}_k(\boldsymbol{x_\delta})$ w.r.t. $\boldsymbol{\rho}_{\texttt{txt}}, \boldsymbol{\rho}_{\texttt{img}}$ and $\boldsymbol{f}_k(\boldsymbol{x})$ | single/ multiple objects |

## 2 Related works

**Adversarial attacks on classifiers.** Several state-of-the-art adversarial attacks [2, 6, 8–17, 52–60] have been designed to disturb the predictions of classifiers. Broadly these approaches can be categorized into two strategies: instance (or image) specific attacks and generative model-based attacks. Instance specific attacks [2, 6, 8, 9, 14–17, 52–60] create perturbations for every image exclusively. Specifically, these perturbations are computed by querying the victim model for multiple iterations in order to eventually alter the image imperceptibly (e.g. texture level changes to image [60]) to cause its misclassification. Due to this "specific to image" strategy, their time-complexity to alter the decision of a large set of images has been shown to be extremely poor [11, 13, 31]. Furthermore, learning perturbations based on single-image generally restrict their success of misclassification only on the known models [11, 13].

To alleviate these drawbacks, a new category of attack strategies has been explored in [10–13, 61] where a generative model is adversarially trained against a surrogate victim model (in other words, treated as a *discriminator*) to craft perturbations on whole data distribution. This attack strategy particularly allows one to perturb multiple images simultaneously once the generative model is optimized, as well as enhances the transferability of perturbations to unseen black-box models [10, 11]. For example, Generative Adversarial Perturbations or GAP [10] and Cross-Domain Attack or CDA [11] presented a distribution-driven attack that trains a generative model for creating adversarial examples by utilizing the cross-entropy loss and relativistic cross-entropy loss [62] objective, respectively. Different from these, Transferable Adversarial Perturbations or TAP [12] and Beyond ImageNet Attack or BIA [13] presented an attack methodology to further enhance transferability of perturbations using feature separation loss functions (*e.g.* mean square error loss) at mid-level layers of the surrogate model. Most of these methods focused on creating transferable perturbations assuming that the surrogate model is trained in the same domain as the target victim model [13]. Further, a mid-level layer is manually selected for each architecture and is also sensitive to the dataset (shown later in Section 4). Similarly, [61] proposes to change image attributes to create semantic manipulations using their disentangled representations via generative models. Most of these generative attacks employed classifiers that operate under the regime that input images include single dominant objects. Some recent attacks [26–30] have focused on analyzing complex images which contain multiple objects, however, they are instance-driven attacks that introduce aforesaid drawbacks of transferability and time complexity. In contrast to these aforementioned works, **GAMA** *is a generative model-based attack designed to craft imperceptible adversarial perturbations that can strongly disrupt both multi-label and single-label classifiers.* Moreover, **GAMA** uses a novel perturbation generation strategy that employs a pre-trained CLIP model [36] based framework to craft highly effectual and transferable perturbations by leveraging multi-modal (image and text) embeddings. We summarize the differences between prior generative attacks and **GAMA** in Table 1.

**Applications of Vision-and-Language (VL) representations.** Due to their robust zero-shot performance, joint vision-and-language pre-trained models [35–39] have allowed new language-driven solutions for various downstream tasks [40–51, 63]. The differentiating attribute of using VL models [36], when compared to existing conventional image-based pre-trained models [33], is that they provide high-quality aligned visual and textual representations learnt from large-scale image-text

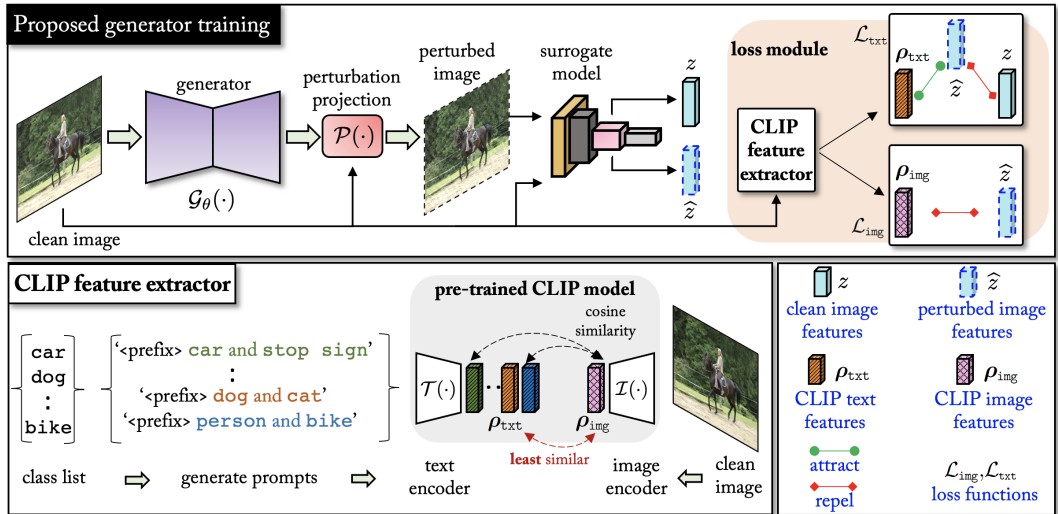

Figure 2: **Overview of GAMA.** The perturbation generator $\mathcal{G}_{\boldsymbol{\theta}}(\cdot)$ crafts a perturbed image ($\ell_\infty$-budget constrained by projection operator $\mathcal{P}(\cdot)$) from the clean image as input. Next, embeddings $\boldsymbol{z}$ from clean image and $\widehat{\boldsymbol{z}}$ from perturbed image are extracted from the surrogate model. A pre-trained CLIP model extracts the image embedding $\boldsymbol{\rho}_{\texttt{img}}$ from the clean image and the text embedding $\boldsymbol{\rho}_{\texttt{txt}}$ that is *least* similar to $\boldsymbol{\rho}_{\texttt{img}}$ (see details in Section 3.1). Finally, the loss functions $\mathcal{L}_{\texttt{img}}$ and $\mathcal{L}_{\texttt{txt}}$ utilize these embeddings to optimize the generator weights $\boldsymbol{\theta}$. Loss solely based on a surrogate model not shown here for simplicity. We use a *prefix*='a photo depicts' in all the text prompts following [67].

pairs. In this work, we leverage one such powerful VL framework named CLIP [36] to an adversary's advantage and show its utility in preparing a perturbation generator for formidable attacks across multiple distributions. Employing freely available pre-trained models for tasks other than what they were trained for has been common practice (*e.g.* VGG [64] models in [65, 66], CLIP for domain adaptation of generators in [46]). To the best of our knowledge, the proposed attack is the first to introduce such VL model usage to subvert classifier decisions.

## 3 Proposed Attack Methodology: GAMA

**Problem Statement.**    Our goal is to train a generative model $\mathcal{G}_{\boldsymbol{\theta}}(\cdot)$ (weights $\boldsymbol{\theta}$) from a training distribution of images with multiple-objects. Once $\boldsymbol{\theta}$ is optimized, $\mathcal{G}_{\boldsymbol{\theta}}(\cdot)$ can create perturbations on diverse types (multi-object or otherwise) of input images that can lead to misclassification on an unknown victim classifier. Suppose we have access to a *source* dataset $\mathcal{D}$ consisting of $N$ training samples from $C$ number of classes, with each sample/image possibly consisting of multiple object labels, i.e., multi-label images. Each $i^{th}$ sample in $\mathcal{D}$ is represented as $\boldsymbol{x}^{(i)} \in \mathbb{R}^{H \times W \times T}$ (with height $H$, width $W$, and channels $T$) containing labels $\boldsymbol{y}^{(i)} = [y_1^{(i)}, \cdots, y_C^{(i)}] \in \mathcal{Y} \subseteq \{0,1\}^C$. More specifically, if sample $\boldsymbol{x}^{(i)}$ is associated with class $c$, $y_c^{(i)} = 1$ indicates the existence of an object from class $c$ in $\boldsymbol{x}^{(i)}$. Further, we have access to a surrogate multi-label classifier trained on $\mathcal{D}$ denoted as $\boldsymbol{f}(\cdot)$ which is employed to optimize the perturbation generator $\mathcal{G}_{\boldsymbol{\theta}}(\cdot)$'s weight $\boldsymbol{\theta}$. For ease of exposition, we drop the superscript $i$ in further discussion.

### 3.1 Adversary Equipped with Pre-Trained CLIP

We aim to train a generator $\mathcal{G}_{\boldsymbol{\theta}}(\cdot)$ that learns to create perturbations from its observations by fooling a surrogate classifier $\boldsymbol{f}(\cdot)$ during its training phase. Now, as $\mathcal{G}_{\boldsymbol{\theta}}(\cdot)$ learns to create perturbations $\boldsymbol{\delta}$ in accordance to $\boldsymbol{f}(\cdot)$, it is bounded by the features extracted from $\boldsymbol{f}(\cdot)$ in order to contrast $\boldsymbol{x}$ and $\boldsymbol{x}_\delta$ (*e.g.* final-layer logits in [10, 11] or mid-level features [12, 13]). In this work, we explore a case where we have access to a pre-trained vision-and-language model like CLIP that can be utilized as a loss network to train $\mathcal{G}_{\boldsymbol{\theta}}(\cdot)$. Our motivation for using CLIP is to exploit its joint text and image matching property and compute two embeddings: clean image embedding extracted from the image encoder and a *dissimilar* text embedding extracted from the text encoder. Specifically, we aim to encode

the contextual relationships between multiple objects in the natural scene via language derivatives. We next describe **GAMA**'s method and present a novel strategy to use CLIP's model to train $\mathcal{G}_{\boldsymbol{\theta}}(\cdot)$. Note that we assume each image contains two co-occurring classes for creating text prompts, mainly restricted due to computation of co-occurrence matrices of dimension $C \times C$ available for multi-label datasets. As we will see later, co-occurrence matrices allow us to discard pairs of classes that would not occur in real-world scenarios.

**GAMA Overview.** Before training $\mathcal{G}_{\boldsymbol{\theta}}(\cdot)$, we first compute a text embedding matrix $\boldsymbol{A}_{\texttt{txt}} = [\boldsymbol{\rho}_1, \boldsymbol{\rho}_2, \cdots, \boldsymbol{\rho}_N] \in \mathbb{R}^{N \times K}$ with $\boldsymbol{\rho}_n \in \mathbb{R}^K$ (explained in detail later) using the CLIP text encoder $\mathcal{T}(\cdot)$. Here, $K$ is the embedding size of output from $\mathcal{T}(\cdot)$. During training $\mathcal{G}_{\boldsymbol{\theta}}(\cdot)$, we start by feeding the clean image $\boldsymbol{x}$ to the CLIP image encoder $\mathcal{I}(\cdot)$ and computing an image embedding $\boldsymbol{\rho}_{\texttt{img}} = \mathcal{I}(\boldsymbol{x}) \in \mathbb{R}^K$. Next, a particular vector $\boldsymbol{\rho}_{\texttt{txt}} \in \mathbb{R}^K$ from $\boldsymbol{A}_{\texttt{txt}}$ is retrieved that is least similar to $\boldsymbol{\rho}_{\texttt{img}}$. Then, we feed $\boldsymbol{x}$ to $\mathcal{G}_{\boldsymbol{\theta}}(\cdot)$ and create $\boldsymbol{x}_{\boldsymbol{\delta}}$ while ensuring it to be under given perturbation $\ell_{\infty}$ budget $\epsilon$ using the perturbation projection operator $\mathcal{P}(\cdot)$. These clean and perturbed images are then fed to the surrogate classifier $\boldsymbol{f}(\cdot)$ to extract $K$-dimensional embeddings at specific $k$th layer, denoted by $\boldsymbol{f}_k(\boldsymbol{x})$ and $\boldsymbol{f}_k(\boldsymbol{x}_{\boldsymbol{\delta}})$ respectively. Finally, the aforementioned quadruplet embeddings $(\boldsymbol{\rho}_{\texttt{txt}}, \boldsymbol{\rho}_{\texttt{img}}, \boldsymbol{f}_k(\boldsymbol{x}), \boldsymbol{f}_k(\boldsymbol{x}_{\boldsymbol{\delta}}))$ are used to compute a contrastive learning based CLIP text embedding-guided loss $\mathcal{L}_{\texttt{txt}}(\boldsymbol{\rho}_{\texttt{txt}}, \boldsymbol{f}_k(\boldsymbol{x}), \boldsymbol{f}_k(\boldsymbol{x}_{\boldsymbol{\delta}}))$ and a regression learning based CLIP image embedding-guided loss $\mathcal{L}_{\texttt{img}}(\boldsymbol{\rho}_{\texttt{img}}, \boldsymbol{f}_k(\boldsymbol{x}_{\boldsymbol{\delta}}))$ to compute the final objective $\mathcal{L}$. We also include a loss function that further maximizes the difference between $\boldsymbol{f}_k(\boldsymbol{x})$ and $\boldsymbol{f}_k(\boldsymbol{x}_{\boldsymbol{\delta}})$ solely from the surrogate classifier's perspective. This loss $\mathcal{L}$ is minimized to update the weights of the generator $\boldsymbol{\theta}$. The whole **GAMA** paradigm is illustrated in Figure 2 and summarized in Algorithm 1. The details of loss objectives $\mathcal{L}_{\texttt{img}}$ and $\mathcal{L}_{\texttt{txt}}$ (with text embedding matrix $\boldsymbol{A}_{\texttt{txt}}$) are discussed next.

**CLIP text embedding-guided loss ($\mathcal{L}_{\texttt{txt}}$).** Let $\boldsymbol{z} = \boldsymbol{f}_k(\boldsymbol{x})$ and $\widehat{\boldsymbol{z}} = \boldsymbol{f}_k(\boldsymbol{x}_{\boldsymbol{\delta}})$. The CLIP framework inherently learns the text and vision embedding association via a contrastive learning regime [36, 68], constraining the feature embeddings of the input image and its counterpart language description to be as similar as possible. Different from CLIP's image embedding $\boldsymbol{\rho}_{\texttt{img}}$, CLIP's text embedding $\boldsymbol{\rho}_{\texttt{txt}}$ allows us to look beyond the pixel-based features. More specifically, CLIP's vision-and-language aligning ability allows us to utilize *text* features to craft transferable image perturbations. Hence, we can optimize $\mathcal{G}_{\boldsymbol{\theta}}(\cdot)$ to create perturbed images $\boldsymbol{x}_{\boldsymbol{\delta}}$ that do not follow the same text embedding alignment as their clean image counterpart $\boldsymbol{x}$. In order to cause this text misalignment, we create a triplet of embeddings where the anchor $\widehat{\boldsymbol{z}}$ is pushed away from $\boldsymbol{z}$ while pulling it closer to a text embedding $\boldsymbol{\rho}_{\texttt{txt}}$ that is least associated or similar to a clean image $\boldsymbol{x}$. To compute this triplet, the following two steps are performed.

- **Before training, compute $\boldsymbol{A}_{\texttt{txt}}$.** The goal is create a dictionary or matrix of text embeddings which can be utilized to retrieve $\boldsymbol{\rho}_{\texttt{txt}}$ during optimization of $\mathcal{G}_{\boldsymbol{\theta}}(\cdot)$. Firstly, we generate language derivatives or *text prompts* using classes of source distribution. This means we only need to know all the available $C$ classes in $\mathcal{D}$ but not their specific association with $\boldsymbol{x}$. Secondly, with assumption that each clean image $\boldsymbol{x}$ is associated with two classes, we can generate $C^2$ text prompts and create a matrix $\boldsymbol{A}_{\texttt{txt}}$ of size $C^2 \times K$. For example, if classes 'cat', 'dog', 'person' and 'boat' exist in $\mathcal{D}$, then one can create text prompts such as "a photo depicts cat and dog" or "a photo depicts person and boat" (see Figure 1 for 10 random examples extracted from CLIP's 'ViT-B/16' model using Pascal-VOC's classes). Here, the part of the text prompt underlined is a recommended 'prefix' common to all text prompts as suggested in [67]. However, such $\boldsymbol{A}_{\texttt{txt}}$ can contain embeddings from prompts that are generated from classes that do not exist in real life. To circumvent this, we utilize an object co-occurrence matrix $\mathcal{O} \in \mathbb{R}^{C \times C}$ (a binary matrix) to estimate the co-occurrence relationships between classes. Computed from the training data set containing $C$ classes, $\mathcal{O}$ is first initialized with a matrix containing only zeros. Then, an element $\mathcal{O}_{ij}$ ($i$th row and $j$th column of $\mathcal{O}$) is set to 1 if objects from classes $y_i$ and $y_j$ appear together at least in one image. Computing such co-occurrence information is a common practice in multi-object downstream problems [26, 27, 69–72]. We use $\mathcal{O}$ provided by [69]. Using such a co-occurrence matrix, we only create text prompts from a pair of classes that occur together according to $\mathcal{O}$. This leads to a text embedding matrix of size $\boldsymbol{A}_{\texttt{txt}}$ of size $\|\mathcal{O}\|_0 \times K$ where $\|\mathcal{O}\|_0$ denotes total non-zero elements.

- **During training, compute $\boldsymbol{\rho}_{\texttt{txt}}$.** CLIP's training objective allows it to push the embeddings of associated image-text pairs closer compared to non-matched pairs. We leverage this property to compute the least similar text embedding $\boldsymbol{\rho}_{\texttt{txt}}$ w.r.t. image embedding $\boldsymbol{\rho}_{\texttt{img}}$. During each training

epoch, we randomly sample $B$ candidates $[\boldsymbol{\rho}_1, \boldsymbol{\rho}_2, \cdots, \boldsymbol{\rho}_B]$ from $\boldsymbol{A}_{\text{txt}}$ and estimate $\boldsymbol{\rho}_{\text{txt}}$ as follows:

$$\boldsymbol{\rho}_{\text{txt}} = \min[\text{cs}(\boldsymbol{\rho}_{\text{img}}, \boldsymbol{\rho}_1), \text{cs}(\boldsymbol{\rho}_{\text{img}}, \boldsymbol{\rho}_2), \cdots, \text{cs}(\boldsymbol{\rho}_{\text{img}}, \boldsymbol{\rho}_B)] \tag{1}$$

Here, $\text{cs}(\cdot)$ denotes cosine similarity. Next, we force $\widehat{\boldsymbol{z}}$ to align with $\boldsymbol{\rho}_{\text{txt}}$ while misaligning with $\boldsymbol{z}$. This is implemented as contrastive learning [73, 74] objective as follows.

$$\mathcal{L}_{\text{txt}} = \min_{\boldsymbol{\theta}} \; {}^{1}\!/_{K}\Big( \|\widehat{\boldsymbol{z}} - \boldsymbol{\rho}_{\text{txt}}\|_2^2 + \big[\alpha - \|\widehat{\boldsymbol{z}} - \boldsymbol{z}\|_2\big]_+ \Big) \tag{2}$$

where $\alpha > 0$ is the desired margin between clean and perturbed image embedding, and $[v]_+ = \max(0, v)$. $\mathcal{L}_{\text{txt}}$ pulls away embeddings of $\boldsymbol{x}$ and $\boldsymbol{x}_\delta$ by making them keep a margin $\alpha$ while pushing dissimilar embeddings $\widehat{\boldsymbol{z}}$ and $\boldsymbol{\rho}_{\text{txt}}$ closer than the given margin.

**CLIP image embedding-guided loss ($\mathcal{L}_{\text{img}}$).** Due to CLIP's learning on $\sim$400 million internet retrieved images from diverse categories and its consequential strong zero-shot image recognition performance over different distributions [36, 41], we argue that its image encoder $\mathcal{I}(\cdot)$ outputs an embedding that has captured attributes of input image with distinct generalized visual features. **GAMA** leverages this to our advantage, and maximizes the difference between $\widehat{\boldsymbol{z}}$ and CLIP's image encoder's embedding for clean image $\boldsymbol{\rho}_{\text{img}}$. The aim of such an objective is to increase the transferability strength of $\mathcal{G}_{\boldsymbol{\theta}}(\cdot)$ perturbations using the generalized features computed from $\mathcal{I}(\cdot)$. This is envisioned using a regression learning based loss described as follows:

$$\mathcal{L}_{\text{img}} = \min_{\boldsymbol{\theta}} \; -\big({}^{1}\!/_{K}\|\boldsymbol{\rho}_{\text{img}} - \widehat{\boldsymbol{z}}\|_2^2\big) \tag{3}$$

**Final Learning Objective ($\mathcal{L}$).** Loss functions $\mathcal{L}_{\text{img}}$ and $\mathcal{L}_{\text{txt}}$ are finally added to a surrogate model loss $\mathcal{L}_{\text{surr}}$ that minimizes the cosine similarity of $\boldsymbol{z}$ and $\widehat{\boldsymbol{z}}$ [13]. Choice of layer $k$ is dependent on feature outputs of the CLIP model employed. All embeddings are normalized before computing the loss functions.

$$\mathcal{L} = \min_{\boldsymbol{\theta}} \; \big(\mathcal{L}_{\text{surr}} + \mathcal{L}_{\text{img}} + \mathcal{L}_{\text{txt}}\big) \tag{4}$$

Overall, $\mathcal{L}_{\text{surr}}$ maximizes the difference between $\boldsymbol{x}$ and $\boldsymbol{x}_\delta$ from surrogate $\boldsymbol{f}(\cdot)$'s perspective, while $\mathcal{L}_{\text{img}}$ and $\mathcal{L}_{\text{txt}}$ enhance its transferability using CLIP's perspective.

**Attack evaluation.** We assume that the attacker has no knowledge of victim classifier $\boldsymbol{g}(\cdot)$ and its data distribution $\mathcal{D}_t$. Further, there is a perturbation budget of $\epsilon$ defined by an $\ell_\infty$ norm. To launch an attack, we input a clean image $\boldsymbol{x}_t$ from target dataset $\mathcal{D}_t$ to optimized $\mathcal{G}_{\boldsymbol{\theta}}(\cdot)$ and craft imperceptible perturbations $\boldsymbol{\delta}_t$ in order to alter the decision space of the target victim classifier $\boldsymbol{g}(\cdot)$ (pre-trained on $\mathcal{D}_t$). Mathematically, this can be represented as $\boldsymbol{y}_t \neq \widehat{\boldsymbol{y}}_t$ where, $\boldsymbol{y}_t = \boldsymbol{g}(\boldsymbol{x}_t)$ and $\widehat{\boldsymbol{y}}_t = \boldsymbol{g}(\boldsymbol{x}_t + \boldsymbol{\delta}_t)$ with $\|\boldsymbol{\delta}_t\|_\infty \leq \epsilon$. We can cause following attack scenarios after training $\mathcal{G}_{\boldsymbol{\theta}}(\cdot)$ against $\boldsymbol{f}(\cdot)$ on $\mathcal{D}$:

- *Scenario 1*: an attack termed *white-box* if $\boldsymbol{f}(\cdot) = \boldsymbol{g}(\cdot)$ and $\mathcal{D} = \mathcal{D}_t$
- *Scenario 2*: an attack termed *black-box* if either $\boldsymbol{f}(\cdot) \neq \boldsymbol{g}(\cdot)$ or $\mathcal{D} \neq \mathcal{D}_t$

A real-world attack is generally modeled by *Scenario 2* as an adversary would not have the knowledge of victim model $\boldsymbol{g}(\cdot)$'s architecture, its training data distribution $\mathcal{D}_t$ and the task it performs *e.g.* single-label classification, multi-label classification, or object detection, *etc*. The perturbations that make an attack successful in *Scenario 2* should be highly transferable.

## 4 Experiments

In this section, we analyze the strength of **GAMA** under diverse practical attack settings. We also perform an ablation analysis of **GAMA**, test the attack robustness against various defenses ([76, 77], median blurring, context-consistency check), as well performance of attacks on different architecture designs. *Note that* we provide more black-box attack results in the supplementary material.

**Baselines.** As there are no prior works for generative attacks that learn on multi-object scenes using multi-label classifiers, we define our baselines by adapting existing state-of-the-art generative attacks summarized in Table 1. Specifically, the cross-entropy loss in GAP [10] and CDA [11] is replaced with binary cross-entropy loss to handle the prediction of multiple labels during training.

**Training Details.** We use the multi-label datasets PASCAL-VOC [78] and MS-COCO [79] to train

---

**Algorithm 1: GAMA** pseudo-code

---

   **Input** : distribution $\mathcal{D}$, batch size $B$, perturbation $\ell_\infty$ bound $\epsilon$
   **Input** : surrogate classifier $\boldsymbol{f}(\cdot)$, CLIP-encoders for text $\mathcal{T}(\cdot)$ and image $\mathcal{I}(\cdot)$
   **Output** : optimized perturbation generator $\mathcal{G}_{\boldsymbol{\theta}}(\cdot)$'s weights $\boldsymbol{\theta}$

---

1   Randomly initialize $\boldsymbol{\theta}$. Load (as well as freeze) $\boldsymbol{f}(\cdot)$, $\mathcal{T}(\cdot)$ and $\mathcal{I}(\cdot)$ with respective pre-trained weights
2   Create text embeddings matrix $\boldsymbol{A}_{\texttt{txt}}$ from $\mathcal{T}(\cdot)$ as described in Section 3.1
3   **repeat**
4      Input $\boldsymbol{x}$ to $\mathcal{I}(\cdot)$ and get $\boldsymbol{\rho}_{\texttt{img}}$
5      Randomly sample $B$ vectors from $\boldsymbol{A}_{\texttt{txt}}$ and get least similar text embedding $\boldsymbol{\rho}_{\texttt{txt}}$ w.r.t. $\boldsymbol{\rho}_{\texttt{img}}$
6      Input clean image $\boldsymbol{x}$ (from $\mathcal{D}$) to $\boldsymbol{f}(\cdot)$ and compute mid-level embedding $\boldsymbol{f}_k(\boldsymbol{x})$
7      Input $\boldsymbol{x}$ to $\mathcal{G}_{\boldsymbol{\theta}}(\cdot)$ and project it within bound $\epsilon$ using $\mathcal{P}(\cdot)$ to obtain $\boldsymbol{x}_{\boldsymbol{\delta}}$
8      Input $\boldsymbol{x}_{\boldsymbol{\delta}}$ to $\boldsymbol{f}(\cdot)$ and compute mid-level embedding $\boldsymbol{f}_k(\boldsymbol{x}_{\boldsymbol{\delta}})$
9      Compute loss $\mathcal{L}$ by Equation (4) and minimize it to update $\boldsymbol{\theta}$ using Adam [75]
10   **until** *convergence*

---

generators for the baselines and our method. Unless otherwise stated, perturbation budget is set to $\ell_\infty \leq 10$ for all experiments. We chose the following surrogate models $\boldsymbol{f}(\cdot)$ (Pascal-VOC or MS-COCO pre-trained multi-label classifiers): ResNet152 (Res152) [80], DenseNet169 (Den169) [81], and VGG19 [64]. For the CLIP model, we use the 'ViT-B/16' framework [36]. See supplementary material for more training details.

**Inference Metrics.** We measure attack performances on multi-label classifiers using hamming score (%) defined in [82, 83]. For evaluations on single-label classifiers and object detectors, we use top-1 accuracy (%) and `bbox_mAP_50` $\in [0, 1]$ metric, respectively. A lower score indicates better attack. Best results are in **bold**. For reference, accuracy on clean images is provided as 'No Attack'.

## 4.1 Results and Analysis

All trained perturbation generators (trained only on multi-label datasets) are extensively evaluated under following victim model settings.

• *White-box and black-box (multi-label classification, different model than $\boldsymbol{f}(\cdot)$):* We evaluate the attacks in white-box and black-box settings on six victim multi-label classifiers (VGG16, VGG19, ResNet50 (Res50), Res152, Den169, and DenseNet121 (Den121)) in Table 2 and Table 3 (white-box attacks are marked with cell color). We outperform all baselines in the majority of cases, with an average absolute difference (w.r.t. closest method) of ~13 percentage points (pp) for Pascal-VOC and ~4.46pp for MS-COCO.

• *Black-box (single-label classification):* We evaluate the attacks in a black-box setting with various single-label classifiers for CIFAR10/100 [84] (coarse-grained tasks [13]), CUB-200-2011 (CUB) [32], Stanford Cars (Car) [85], and FGVC Aircrafts (Air) [86] (fine-grained tasks [13]) in Table 6, and ImageNet [87] (50K validation set) in Table 4 and Table 5. Following [13], the victim models of coarse-grained tasks are taken from [88], fine-grained task models (Res50, SENet154 (SeNet), and SE-ResNet101 (se-Res101) [89]) from [90], and six ImageNet models from [33]. Here, we beat our closest baseline in all cases by ~13.33pp for Pascal-VOC and ~5.83pp for MS-COCO on six ImageNet models. Note that the ImageNet results also demonstrate the drop in performance of TAP [12] and BIA [13] attacks that show close to 0% top-1 accuracy when $\mathcal{G}_{\boldsymbol{\theta}}(\cdot)$ is trained on ImageNet on the attacker side [12, 13]. We hypothesize that such a drop in performance is due to sensitivity to the dataset of the manually selected mid-level layer of $\boldsymbol{f}(\cdot)$ used by the attacker. We observe a similar trend when attacking non-ImageNet distributions as suggested by BIA [13] in coarse and fine-grained tasks in Table 6. In this case, **GAMA** beats the prior attacks by average ~13.33pp when $\mathcal{G}_{\boldsymbol{\theta}}(\cdot)$ is trained with Pascal-VOC.

• *Black-box (Object detection):* We also evaluate a difficult black-box attack with state-of-the-art MS-COCO object detectors (Faster RCNN with Res50 backbone (FRCN) [91], RetinaNet with Res50 backbone (RNet) [92], DEtection TRansformer (DETR) [93], and Deformable DETR ($D^2$ETR) [94]) in Table 7, available from [95]. It can be observed that **GAMA** outperforms its competitors when $\mathcal{G}_{\boldsymbol{\theta}}(\cdot)$ is trained with Pascal-VOC.

Table 2: Pascal-VOC → Pascal-VOC

| $f(\cdot)$ | Method | VGG16 | VGG19 | Res50 | Res152 | Den169 | Den121 | Average |
|---|---|---|---|---|---|---|---|---|
| | No Attack | 82.51 | 83.18 | 80.52 | 83.12 | 83.74 | 83.07 | 82.69 |
| VGG19 | GAP [10] | 19.64 | 16.60 | 72.95 | 76.24 | 68.79 | 66.50 | 53.45 |
| | CDA [11] | 26.16 | 20.52 | 61.40 | 65.67 | 70.33 | 62.67 | 51.12 |
| | TAP [12] | 24.77 | 19.26 | 66.95 | 66.95 | 68.65 | 64.51 | 51.84 |
| | BIA [13] | 12.53 | 14.00 | 64.24 | 69.07 | 69.44 | 64.71 | 48.99 |
| | GAMA | 6.11 | 5.89 | 41.17 | 45.57 | 53.11 | 44.58 | 32.73 |
| Res152 | GAP [10] | 56.93 | 56.20 | 65.58 | 72.26 | 75.22 | 69.54 | 65.95 |
| | CDA [11] | 41.07 | 47.60 | 53.84 | 47.22 | 67.50 | 59.65 | 52.81 |
| | TAP [12] | 52.92 | 58.24 | 56.52 | 53.61 | 71.55 | 64.56 | 59.56 |
| | BIA [13] | 45.34 | 49.74 | 51.98 | 50.27 | 67.75 | 61.05 | 54.35 |
| | GAMA | 33.42 | 39.42 | 32.39 | 20.46 | 49.76 | 49.54 | 37.49 |
| Den169 | GAP [10] | 62.09 | 59.55 | 68.60 | 72.81 | 76.09 | 72.70 | 68.64 |
| | CDA [11] | 52.28 | 53.75 | 59.65 | 67.23 | 69.60 | 67.37 | 61.64 |
| | TAP [12] | 58.48 | 58.55 | 58.14 | 63.42 | 52.66 | 62.57 | 58.97 |
| | BIA [13] | 48.52 | 53.77 | 56.15 | 63.33 | 54.01 | 58.85 | 55.77 |
| | GAMA | 44.25 | 52.89 | 48.83 | 53.25 | 45.00 | 50.96 | 49.19 |

Table 3: MS-COCO → MS-COCO

| $f(\cdot)$ | Method | VGG16 | VGG19 | Res50 | Res152 | Den169 | Den121 | Average |
|---|---|---|---|---|---|---|---|---|
| | No Attack | 65.80 | 66.48 | 65.64 | 67.95 | 67.59 | 66.39 | 66.64 |
| VGG19 | GAP [10] | 8.31 | 10.61 | 39.49 | 48.00 | 41.00 | 38.12 | 30.92 |
| | CDA [11] | 6.57 | 8.57 | 37.38 | 43.56 | 38.41 | 35.59 | 28.34 |
| | TAP [12] | 3.45 | 6.14 | 25.77 | 29.56 | 20.05 | 21.15 | 17.68 |
| | BIA [13] | 2.47 | 4.01 | 30.76 | 37.34 | 26.40 | 27.95 | 21.48 |
| | GAMA | 3.59 | 3.75 | 27.13 | 30.43 | 24.60 | 21.77 | 18.54 |
| Res152 | GAP [10] | 42.59 | 45.41 | 51.22 | 53.75 | 54.18 | 52.54 | 49.94 |
| | CDA [11] | 30.16 | 37.79 | 42.83 | 45.13 | 49.24 | 44.93 | 41.68 |
| | TAP [12] | 24.34 | 25.94 | 29.40 | 24.13 | 35.58 | 33.06 | 28.74 |
| | BIA [13] | 22.73 | 22.76 | 28.64 | 22.16 | 36.06 | 32.41 | 27.46 |
| | GAMA | 24.52 | 27.73 | 30.62 | 23.04 | 31.30 | 27.31 | 27.42 |
| Den169 | GAP [10] | 29.85 | 32.77 | 38.15 | 40.84 | 24.98 | 33.99 | 33.43 |
| | CDA [11] | 39.39 | 41.19 | 46.34 | 50.82 | 43.42 | 44.63 | 44.29 |
| | TAP [12] | 23.01 | 27.73 | 32.75 | 40.22 | 15.73 | 20.90 | 26.72 |
| | BIA [13] | 27.01 | 29.59 | 34.65 | 43.42 | 13.57 | 24.69 | 28.82 |
| | GAMA | 10.40 | 13.47 | 19.30 | 23.46 | 8.65 | 10.29 | 14.26 |

Table 4: Pascal-VOC → ImageNet

| $f(\cdot)$ | Method | VGG16 | VGG19 | Res50 | Res152 | Den121 | Den169 | Average |
|---|---|---|---|---|---|---|---|---|
| | No Attack | 70.15 | 70.94 | 74.60 | 77.34 | 74.22 | 75.74 | 73.83 |
| VGG19 | GAP [10] | 24.44 | 21.64 | 63.65 | 67.84 | 63.09 | 65.47 | 51.02 |
| | CDA [11] | 13.83 | 11.99 | 47.32 | 53.92 | 46.81 | 52.24 | 37.68 |
| | TAP [12] | 06.70 | 07.28 | 50.94 | 57.36 | 47.68 | 53.43 | 37.23 |
| | BIA [13] | 04.20 | 04.73 | 48.63 | 57.65 | 45.94 | 53.37 | 35.75 |
| | GAMA | 03.07 | 03.41 | 22.32 | 34.04 | 24.51 | 30.35 | 19.61 |
| Res152 | GAP [10] | 34.04 | 34.67 | 52.85 | 61.61 | 58.09 | 59.24 | 50.08 |
| | CDA [11] | 29.33 | 34.88 | 44.28 | 46.05 | 46.91 | 51.62 | 42.17 |
| | TAP [12] | 33.25 | 37.53 | 41.18 | 42.14 | 50.96 | 56.45 | 43.58 |
| | BIA [13] | 22.82 | 27.44 | 34.66 | 36.74 | 45.48 | 51.26 | 36.40 |
| | GAMA | 16.43 | 17.02 | 21.93 | 17.07 | 31.63 | 30.57 | 22.44 |
| Den169 | GAP [10] | 42.79 | 45.01 | 57.79 | 65.42 | 63.02 | 65.31 | 56.55 |
| | CDA [11] | 36.67 | 37.51 | 52.30 | 61.78 | 54.68 | 57.85 | 50.13 |
| | TAP [12] | 28.92 | 30.19 | 38.36 | 50.92 | 45.88 | 40.78 | 39.17 |
| | BIA [13] | 26.12 | 27.42 | 37.06 | 51.30 | 40.63 | 37.56 | 36.68 |
| | GAMA | 18.16 | 20.93 | 28.04 | 41.85 | 26.11 | 21.67 | 26.12 |

Table 5: MS-COCO → ImageNet

| $f(\cdot)$ | Method | VGG16 | VGG19 | Res50 | Res152 | Den121 | Den169 | Average |
|---|---|---|---|---|---|---|---|---|
| | No Attack | 70.15 | 70.94 | 74.60 | 77.34 | 74.22 | 75.74 | 73.83 |
| VGG19 | GAP [10] | 15.55 | 15.06 | 49.50 | 56.07 | 47.65 | 53.49 | 39.55 |
| | CDA [11] | 13.05 | 12.59 | 46.77 | 52.58 | 43.55 | 50.03 | 36.42 |
| | TAP [12] | 02.33 | 02.93 | 19.28 | 35.20 | 19.45 | 23.42 | 17.10 |
| | BIA [13] | 02.51 | 03.09 | 29.72 | 43.98 | 30.37 | 36.53 | 24.36 |
| | GAMA | 02.01 | 02.57 | 19.99 | 35.21 | 26.26 | 32.98 | 19.83 |
| Res152 | GAP [10] | 22.98 | 24.41 | 32.74 | 32.35 | 39.56 | 44.11 | 32.69 |
| | CDA [11] | 35.69 | 39.40 | 51.75 | 54.84 | 53.55 | 58.92 | 49.02 |
| | TAP [12] | 13.29 | 12.46 | 23.44 | 21.11 | 35.14 | 41.29 | 24.45 |
| | BIA [13] | 14.98 | 14.98 | 25.40 | 21.98 | 34.11 | 37.62 | 24.84 |
| | GAMA | 17.94 | 19.16 | 24.57 | 17.24 | 29.67 | 30.57 | 23.19 |
| Den169 | GAP [10] | 30.50 | 30.79 | 40.82 | 51.12 | 41.03 | 37.46 | 38.62 |
| | CDA [11] | 35.75 | 36.69 | 50.45 | 57.43 | 51.23 | 52.44 | 47.33 |
| | TAP [12] | 21.45 | 26.45 | 27.30 | 45.76 | 30.83 | 25.34 | 29.52 |
| | BIA [13] | 20.91 | 25.01 | 37.16 | 50.65 | 34.71 | 23.38 | 31.97 |
| | GAMA | 06.94 | 10.63 | 10.97 | 21.60 | 13.92 | 08.22 | 12.04 |

Table 6: Pascal-VOC → Coarse (CIFAR10/100) and Fine-grained (CUB, Car, Air) tasks

| $f(\cdot)$ | Method | CIFAR10 [88] | CIFAR100 [88] | CUB Res50 | CUB SeNet | CUB se-Res101 | Car Res50 | Car SeNet | Car se-Res101 | Air Res50 | Air SeNet | Air se-Res101 | Average |
|---|---|---|---|---|---|---|---|---|---|---|---|---|---|
| | No Attack | 93.79 | 74.28 | 87.35 | 86.81 | 86.54 | 94.35 | 93.36 | 92.97 | 92.23 | 92.05 | 91.90 | 89.60 |
| VGG19 | GAP [10] | 73.58 | 39.10 | 78.94 | 79.79 | 80.41 | 82.33 | 85.71 | 87.19 | 81.19 | 81.82 | 79.99 | 77.27 |
| | CDA [11] | 70.40 | 44.68 | 54.76 | 64.74 | 68.99 | 70.87 | 75.64 | 81.78 | 42.87 | 74.38 | 77.20 | 66.02 |
| | TAP [12] | 73.18 | 35.41 | 72.42 | 74.39 | 73.94 | 78.40 | 77.08 | 84.59 | 78.91 | 78.94 | 75.52 | 72.98 |
| | BIA [13] | 59.82 | 27.84 | 68.31 | 65.64 | 73.70 | 75.61 | 67.90 | 81.83 | 75.88 | 66.13 | 76.75 | 67.22 |
| | GAMA | 53.85 | 24.94 | 53.52 | 62.19 | 66.93 | 60.08 | 69.11 | 78.95 | 45.51 | 43.71 | 63.37 | 56.56 |
| Res152 | GAP [10] | 69.80 | 41.06 | 64.96 | 80.01 | 81.77 | 72.62 | 86.02 | 87.53 | 84.28 | 84.64 | 85.48 | 76.19 |
| | CDA [11] | 77.60 | 49.43 | 65.38 | 71.52 | 71.63 | 73.04 | 76.52 | 79.54 | 66.61 | 72.73 | 60.10 | 69.46 |
| | TAP [12] | 70.92 | 38.39 | 48.60 | 73.20 | 76.10 | 69.02 | 86.62 | 81.94 | 74.65 | 80.68 | 83.20 | 71.21 |
| | BIA [13] | 67.54 | 36.43 | 51.17 | 70.64 | 71.63 | 70.85 | 82.85 | 80.21 | 72.94 | 80.20 | 81.01 | 69.58 |
| | GAMA | 69.53 | 38.57 | 27.67 | 64.77 | 64.79 | 59.18 | 74.27 | | 59.71 | 69.10 | 65.77 | 61.26 |
| Den169 | GAP [10] | 83.25 | 56.08 | 64.70 | 78.15 | 76.77 | 80.65 | 85.95 | 86.74 | 81.79 | 84.40 | 85.03 | 78.50 |
| | CDA [11] | 84.34 | 58.03 | 61.75 | 73.40 | 71.75 | 84.21 | 85.57 | 84.58 | 78.97 | 82.24 | 78.22 | 76.64 |
| | TAP [12] | 86.77 | 58.67 | 54.04 | 64.45 | 62.31 | 76.13 | 81.35 | 82.91 | 34.02 | 76.66 | 76.75 | 68.55 |
| | BIA [13] | 85.20 | 55.21 | 47.95 | 58.18 | 56.02 | 55.88 | 73.65 | 72.30 | 62.47 | 72.97 | 70.39 | 64.56 |
| | GAMA | 78.27 | 46.80 | 33.57 | 57.44 | 63.24 | 49.31 | 70.65 | 75.14 | 48.48 | 62.95 | 70.15 | 59.63 |

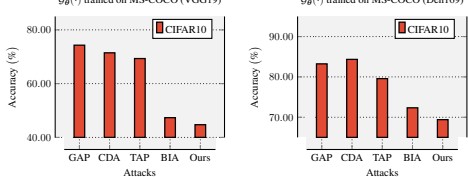
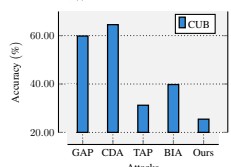
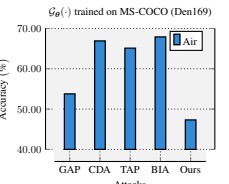

(a) Custom architecture (both from [88])   (b) Standard architecture (*left*: Res50, *right*: se-Res101)

Figure 3: **Transferability on types of victim model designs**. **GAMA** shows potent transferring attacks to victim networks that were custom designed (Figure 3(a)) and that contain standard blocks like Residual blocks [80] (Figure 3(b)).

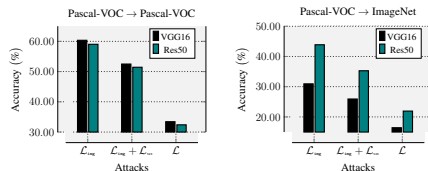

Figure 4: **Ablation analysis of loss objective.** We analyze the contribution due to the introduction of each loss function $\mathcal{L}_{\mathrm{img}}$ and $\mathcal{L}_{\mathrm{txt}}$ towards the final objective $\mathcal{L}$, both in same distribution (*left*) and different distribution (*right*). The surrogate model is Res152.

Table 7: Pascal-VOC → MS-COCO Object Detection task

| $f(\cdot)$ | Method | FRCN | RNet | DETR | D$^2$ETR | Average |
|---|---|---|---|---|---|---|
| | No Attack | 0.582 | 0.554 | 0.607 | 0.633 | 0.594 |
| VGG19 | GAP [10] | 0.424 | 0.404 | 0.360 | 0.410 | 0.399 |
| | CDA [11] | 0.276 | 0.250 | 0.208 | 0.244 | 0.244 |
| | TAP [12] | 0.384 | 0.340 | 0.275 | 0.320 | 0.329 |
| | BIA [13] | 0.347 | 0.318 | 0.253 | 0.281 | 0.299 |
| | **GAMA** | **0.234** | **0.207** | **0.117** | **0.122** | **0.170** |
| Res152 | GAP [10] | 0.389 | 0.362 | 0.363 | 0.408 | 0.380 |
| | CDA [11] | 0.305 | 0.274 | 0.256 | 0.281 | 0.279 |
| | TAP [12] | 0.400 | 0.348 | 0.288 | 0.350 | 0.346 |
| | BIA [13] | 0.321 | 0.275 | 0.205 | 0.256 | 0.264 |
| | **GAMA** | **0.172** | **0.138** | **0.080** | **0.095** | **0.121** |

**Performance on Type of Architectures.** In Figure 3, we further study the transferability of attacks depending on the type of victim architecture: *standard* which follow the standard modules like Residual blocks [80] to build the classifier, and *custom* where the victim classifier doesn't adhere to a specific pattern of network modules. In both cases, **GAMA** consistently maintains better attack rates than other attacks. This shows convincing transferability of perturbations crafted from **GAMA**'s $\mathcal{G}_{\theta}(\cdot)$ under diverse black-box settings. We provide additional results in the supplementary material.

**Robustness of Attacks against Defenses.** To analyze the robustness of all the methods, we launch misclassification attacks ($\mathcal{G}_{\theta}(\cdot)$ trained on MS-COCO with the surrogate model as Den169) when the victim deploys input processing based defense such as median blur with window size as $3 \times 3$, and Neural Representation purifier (NRP) [76] on three ImageNet models (VGG16, Res152, Den169). From Table 8(a) and Table 8(b), we can observe that the attack success of **GAMA** is better than prior methods even when the victim pre-processes the perturbed image before making decisions. In Figure 8(c), we observe that Projected Gradient Descent (PGD) [77] assisted Res50 is difficult to break with **GAMA** performing slightly better than other methods. Finally, motivated by [96], we analyze an output processing defense scenario where the victim can check the context consistency of predicted labels on perturbed images using the co-occurrence matrix $\mathcal{O}$. In particular, if a perturbed image is misclassified showing co-occurrence of labels not present in $\mathcal{O}$, we term this as a *detected attack*. Otherwise, we call it an *undetected attack*. To measure this performance, we first compute the co-occurrence matrix $\mathcal{O}_{\delta}$ by perturbing all the test set images and estimate its precision w.r.t. ground-truth $\mathcal{O}$. To check for attacks that have high precision value $p$ and high misclassification rate, we calculate a 'context score' (higher is better) that is a harmonic mean of $p$ and misclassification rate (1-accuracy). We show the attack performance against this context consistency check in Figure 8(d) for both Pascal-VOC and MS-COCO averaged over all surrogate models under white-box attacks. Clearly, **GAMA** presents itself as the best *undetected* attack compared to prior works.

**Ablation Analysis.** We dissect the contribution of each loss function in our proposed loss objective of Equation (4) in Figure 4 where $\mathcal{G}_{\theta}(\cdot)$ is trained with Pascal-VOC with Res152 surrogate model. We analyze the attack transferability to different victim models (VGG16, Res50). We observe that the introduction of each loss objective (*left* to *right*) increases the strength of the attack both in the same distribution (Pascal-VOC) as the attacker and in the unknown distribution (ImageNet) on both victim classifiers. Finally, we visualize some perturbed image examples crafted by **GAMA** in Figure 5.

| Method | VGG16 | Res152 | Den121 | Average |
|---|---|---|---|---|
| No Attack | 64.57 | 74.04 | 71.68 | 69.92 |
| GAP [10] | 33.33 | 56.90 | 46.34 | 45.52 |
| CDA [11] | 37.89 | 58.98 | 56.19 | 51.02 |
| TAP [12] | 22.37 | 50.67 | 40.81 | 37.95 |
| BIA [13] | 25.09 | 54.45 | 46.34 | 41.96 |
| **GAMA** | **20.34** | **49.66** | **37.55** | **35.85** |

(a) Median Blur

| Method | VGG16 | Res152 | Den121 | Average |
|---|---|---|---|---|
| No Attack | 56.26 | 62.37 | 68.62 | 62.41 |
| GAP [10] | 31.08 | 45.11 | 37.85 | 38.01 |
| CDA [11] | 34.61 | 47.64 | 51.32 | 44.52 |
| TAP [12] | 20.06 | 36.54 | 19.70 | 25.43 |
| BIA [13] | 19.94 | 41.03 | 20.07 | 23.68 |
| **GAMA** | **7.38** | **19.00** | **7.87** | **11.41** |

(b) NRP

(c) PGD ($\epsilon = 4$)

(d) Context check

Table 8: **Robustness Analysis against various defenses**. Our proposed attack **GAMA** consistently shows better performances compared to baselines in scenarios where the victim deploys attack defenses.

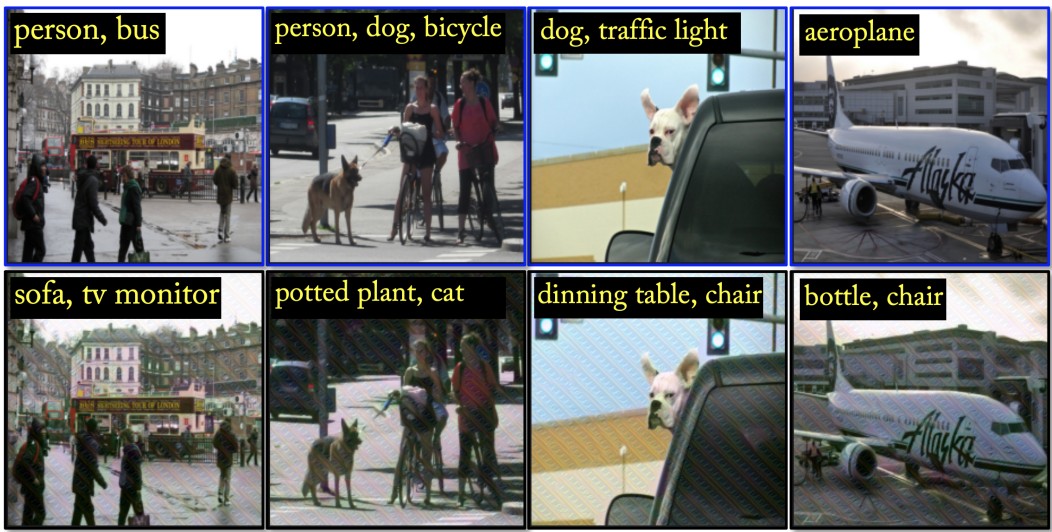

Figure 5: **Qualitative Examples.** We show some clean (*top*) and perturbed (*bottom*) images from **GAMA**. Best viewed in color/zoomed.

## 5 Conclusion

In this paper, we propose a new generative attack **GAMA** that can learn to create perturbations against multi-object scenes. For the first time in generative attack literature, we show the utility of a pre-trained vision-and-language model CLIP to optimize effectual perturbation generators. Specifically, CLIP's joint text and image aligning property allow us to use natural language semantics in order to handle the multi-object semantics in the input scene to be perturbed. To demonstrate **GAMA**'s efficiency, we perform extensive experiments that show state-of-the-art attacks across a wide range of black-box victim models (multi-label/single-label classifiers, and object detectors). We also evaluate the robustness of our attacks against various defense mechanisms. As part of our future works, we will explore more complex methodologies to employ vision-language models both for adversarial attacks and defense systems.

## 6 Limitations and Societal Impacts

**Limitations**. The pre-trained CLIP model 'ViT-B16' outputs a 512-dimensional embedding that restricts us to compare the features extracted from the surrogate model in our losses of the same size. Another limitation of our method is the use of co-occurrence matrices to extract the right pair of classes that exist together in real-world scenes. In this paper, we make an assumption that text prompts are created using two classes that exist together according to the co-occurrence matrix of size $C \times C$ (for $C$ classes in the data distribution). However, we can also use a triplet of classes that exist together in an input scene which would need a co-occurrence tensor of size $C \times C \times C$. Computing such a huge tensor would be tedious to cover all the images provided in the train set (usually in the order of thousands).

**Societal Impacts**. Adversarial attacks are designed with the sole goal to subvert machine decisions by any means available. Our attack approach shows one such method where a benign open-sourced vision-language model can be utilized by an attacker to create potent perturbations. This demonstrates the need for the victim to prepare for constantly evolving attacks that may cause major harm in real-world systems (*e.g.* person re-identification [97]). We believe that our work can help further propagate research into designing efficient and robust models that do not break down to attacks built upon multi-modal (in our case, text-image) features. Future researchers should also be aware of video generative models [98, 99] that can be used to create adversarial attacks for ubiquitous video classifiers built on the success of vision-language models.

**Acknowledgement.** This material is based upon work supported by the Defense Advanced Research Projects Agency (DARPA) under Agreement No. HR00112090096. Approved for public release; distribution is unlimited.

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
