# Supplementary material for "GAMA: Generative Adversarial Multi-Object Scene Attacks"

**Abhishek Aich**,* **Calvin-Khang Ta**\*, **Akash Gupta, Chengyu Song,**
**Srikanth V. Krishnamurthy, M. Salman Asif, Amit K. Roy-Chowdhury**
University of California, Riverside, CA, USA

## CONTENTS

## List of Tables

## List of Figures

---

*Equal contribution. Corresponding author: AA (`aaich001@ucr.edu`). AG is currently with Vimaan AI, USA.

36th Conference on Neural Information Processing Systems (NeurIPS 2022).

We present additional analysis of **GAMA** in the following sections to investigate its attack capabilities under various settings, including black-box embedding visualizations w.r.t. TAP [1], impact of different types of CLIP models, performance with ensemble of surrogate models. We also demonstrate **GAMA**'s transfer attack strength in comparison to prior methods under difficult black-box transfer attacks including in different multi-label distribution, object detection, and robustness of perturbations when victim uses defense mechanisms to minimize classifier performance deterioration. All experiments are done with perturbation budget $\ell_\infty \leq 10$.

# 1 Additional Analysis on GAMA

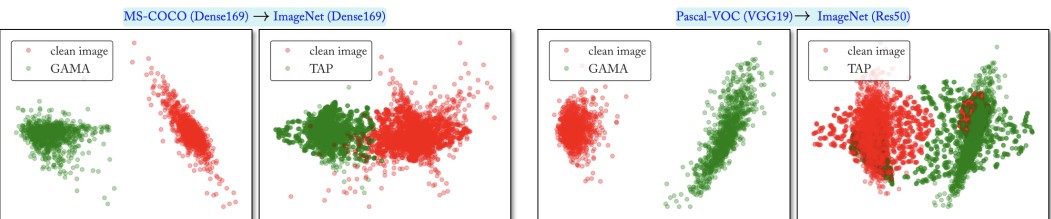

Figure 1: **Embedding visualization. GAMA** uses the CLIP extracted text and image embeddings to craft highly transferable adversarial examples. This can be seen in above embedding visualizations where **GAMA**'s perturbed images lie convincingly farther away from the clean images with better margins compared to TAP [1]. Left and right plots show perturbed image embeddings (both on random 1000 ImageNet images) when $\mathcal{G}_\theta(\cdot)$ is trained with MS-COCO and Pascal-VOC respectively. Surrogate and victim models are given in parenthesis.

**Black-box Setting Embedding Visualization.** To demonstrate the phenomenon that **GAMA** learns to create potent perturbations compared to prior works, we perform Principal Components Analysis (PCA) of perturbed images extracted from **GAMA** and TAP [1] in Figure 1 when the $\mathcal{G}_\theta(\cdot)$ is trained with MS-COCO and Pascal-VOC. We choose PCA visualization as it preserves the global differences of high dimensional data in low dimensional regimes [2, 3]. Clearly, in an unseen distribution (ImageNet [4]), features obtained from **GAMA**'s perturbed images significantly differ from those of clean images in comparison to TAP [1].

**Impact of Different CLIP models.** We analyze the impact of different open-source pre-trained CLIP models provided by Open-AI in Table 1 (surrogate model as Res152), both for the same domain and different domain transfer attacks. We observe that the CLIP frameworks with vision encoders with image transformers [5] (ViT-L/14, ViT-B/32, ViT-B/16) as their backbone perform better in our proposed setting than those with the vision encoders as convolutional networks (RN50, RN101). We attribute this to the effectual representation capability of transformers [5].

Table 1: Impact of CLIP model on **GAMA**

(a) Pascal-VOC $\rightarrow$ Pascal-VOC

|  | VGG16 | VGG19 | Res50 | Res152 | Den169 | Den121 |
|---|---|---|---|---|---|---|
| No Attack | 82.51 | 83.18 | 80.52 | 83.12 | 83.74 | 83.07 |
| RN50 | 8.83 | 15.25 | 64.37 | 67.24 | 70.53 | 69.13 |
| RN101 | 21.74 | 9.45 | 60.56 | 68.53 | 67.01 | 66.17 |
| ViT-L/14 | 43.35 | 49.89 | 45.08 | **43.30** | 54.23 | 51.53 |
| ViT-B/32 | 10.58 | 15.18 | 67.07 | 70.34 | 69.14 | 68.02 |
| ViT-B/16 | **6.12** | **5.89** | **41.17** | 45.57 | **53.11** | **44.58** |

(b) Pascal-VOC $\rightarrow$ ImageNet

|  | VGG16 | VGG19 | Res50 | Res152 | Den121 | Den169 |
|---|---|---|---|---|---|---|
| No Attack | 70.15 | 70.94 | 74.60 | 77.34 | 74.22 | 75.74 |
| RN50 | 3.13 | **2.06** | 46.25 | 52.01 | 49.33 | 45.91 |
| RN101 | 2.93 | 2.41 | 42.73 | 56.16 | 46.67 | 45.97 |
| ViT-L/14 | 16.63 | 20.04 | 26.41 | **23.18** | 31.10 | 32.53 |
| ViT-B/32 | 3.90 | 2.81 | 49.61 | 54.41 | 48.02 | 46.41 |
| ViT-B/16 | **3.07** | 3.41 | **22.32** | 34.04 | **24.51** | **30.35** |

**Random Runs with Error Bars.** We report the mean, and standard error in Table 2 along with the error bar plot (with mean and standard error). We can observe that **GAMA** maintains its performance with random seed values over various runs. Here, $\mathcal{G}_\theta(\cdot)$ was trained on Pascal-VOC with VGG19 as a surrogate.

**Effect of Surrogate Ensemble.** We analyze the results when all the surrogates (VGG19, Res152, and Den169) are employed together to train the perturbation generator $\mathcal{G}_\theta(\cdot)$ using **GAMA**. As can be seen in Table 3 and Table 4 (ensemble denoted as All), we do not observe any significant advantage in results when using multiple surrogates. This same observation has been noted by TAP [1] as well. We

Table 2: Pascal-VOC →Pascal-VOC (s.e. = standard error)

|          | VGG16 | VGG19 | Res50 | Res152 | Den169 | Den121 |
|----------|-------|-------|-------|--------|--------|--------|
| No Attack | 82.51 | 83.18 | 80.52 | 83.12 | 83.74 | 83.07 |
| Run 1    | 5.86  | 5.18  | 45.43 | 50.88  | 52.61  | 43.44  |
| Run 2    | 6.00  | 4.99  | 42.30 | 47.54  | 49.82  | 40.82  |
| Run 3    | 6.08  | 4.88  | 40.71 | 46.64  | 50.31  | 42.73  |
| Run 4    | 5.95  | 5.15  | 42.28 | 45.52  | 51.46  | 40.92  |
| Run 5    | 6.01  | 4.84  | 41.47 | 45.77  | 49.33  | 39.47  |
| mean     | 5.98  | 5.01  | 42.44 | 47.27  | 50.70  | 41.47  |
| s.e.     | 0.035 | 0.067 | 0.800 | 0.966  | 0.590  | 0.711  |

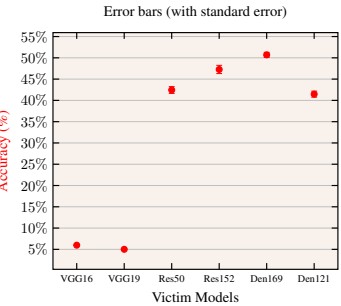

hypothesize that the mid-level features from multiple surrogates may not introduce complementary features to learn comparatively powerful perturbations than single classifier based surrogates.

Table 3: Ensemble comparison: VOC→VOC

| $f(\cdot)$ | VGG16 | VGG19 | Res50 | Res152 | Den169 | Den121 | Average |
|-----------|-------|-------|-------|--------|--------|--------|---------|
| VGG19 | 6.11 | 5.89 | 41.17 | 45.57 | 53.11 | 44.58 | 32.74 |
| Res152 | 33.42 | 39.42 | 32.39 | 20.46 | 49.76 | 49.54 | 37.49 |
| Den169 | 44.25 | 52.89 | 48.83 | 53.25 | 45.00 | 50.96 | 49.19 |
| All | 16.46 | 21.67 | 51.97 | 58.52 | 54.51 | 58.20 | 43.55 |

Table 4: Ensemble comparison: COCO→COCO

| $f(\cdot)$ | VGG16 | VGG19 | Res50 | Res152 | Den169 | Den121 | Average |
|-----------|-------|-------|-------|--------|--------|--------|---------|
| VGG19 | 3.59 | 3.75 | 27.13 | 30.43 | 24.60 | 21.77 | 18.54 |
| Res152 | 24.52 | 27.73 | 30.62 | 23.04 | 31.30 | 27.31 | 27.42 |
| Den169 | 10.40 | 13.47 | 19.30 | 23.46 | 8.65 | 10.29 | 14.26 |
| All | 10.08 | 10.75 | 23.83 | 35.23 | 29.57 | 30.45 | 23.32 |

## 2 Additional Results w.r.t. Baselines

**Black-box Setting (Object Detection).** We evaluate a black-box transfer attack with state-of-the-art MS-COCO object detectors (Faster RCNN with Res50 backbone (FRCN) [9], RetinaNet with Res50 backbone (RNet) [10], DEtection TRansformer (DETR) [11], and Deformable DETR ($D^2$ETR) [12]) in Table 5, provided by [13]. It can be observed that **GAMA** beats the baselines when $\mathcal{G}_{\theta}(\cdot)$ is trained with MS-COCO in the majority of scenarios.

Table 5: COCO → COCO Object Detection

| $f(\cdot)$ | Method | FRCN | RNet | DETR | $D^2$ETR | Average |
|-----------|--------|------|------|------|------|---------|
| | No Attack | 0.582 | 0.554 | 0.607 | 0.633 | 0.594 |
| VGG19 | GAP [6] | 0.347 | 0.312 | 0.282 | 0.304 | 0.311 |
| | CDA [7] | 0.370 | 0.347 | 0.312 | 0.282 | 0.327 |
| | TAP [1] | **0.130** | 0.120 | 0.099 | 0.104 | **0.113** |
| | BIA [8] | 0.266 | 0.229 | 0.185 | 0.211 | 0.223 |
| | **GAMA** | 0.246 | 0.214 | 0.134 | 0.155 | 0.187 |
| Res152 | GAP [6] | 0.187 | 0.145 | 0.097 | 0.108 | 0.134 |
| | CDA [7] | 0.322 | 0.301 | 0.237 | 0.274 | 0.283 |
| | TAP [1] | 0.167 | 0.151 | 0.087 | 0.123 | 0.132 |
| | BIA [8] | **0.152** | 0.144 | 0.101 | 0.121 | 0.129 |
| | **GAMA** | 0.154 | 0.128 | 0.086 | 0.100 | 0.117 |
| Den169 | GAP [6] | 0.308 | 0.261 | 0.201 | 0.213 | 0.245 |
| | CDA [7] | 0.325 | 0.293 | 0.238 | 0.255 | 0.277 |
| | TAP [1] | 0.181 | 0.155 | 0.126 | 0.147 | 0.152 |
| | BIA [8] | 0.265 | 0.236 | 0.185 | 0.214 | 0.225 |
| | **GAMA** | 0.078 | 0.064 | 0.037 | 0.047 | 0.056 |

**Black-box Setting (Multi-Label Classification).** We perform a black-box transfer attack on different multi-label domain than that of $\mathcal{G}_{\theta}(\cdot)$'s training set: Pascal-VOC → MS-COCO in Table 6 and MS-COCO → Pascal-VOC in Table 7. We outperform all baselines in the majority of cases, with an average absolute difference (w.r.t. closest method) of ∼5 percentage points (pp) for Pascal-VOC → MS-COCO, and ∼13.5pp for MS-COCO → Pascal-VOC.

Table 6: Pascal-VOC → MS-COCO

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

(b) NRP

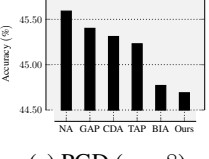

(c) PGD ($\epsilon = 8$)

Table 8: **Robustness Analysis against various defenses**. **GAMA** consistently shows better robustness in cases where victim uses attack defenses ($\mathcal{G}_{\boldsymbol{\theta}}(\cdot)$ trained on Pascal-VOC). 'NA' in Figure 8(c) denotes 'No Attack'.

**Evaluation of adversarial images on CLIP.** We evaluated CLIP (as a "zero-shot prediction" model) on the perturbed images from Pascal-VOC and computed the top two associated labels in Figure 2 using CLIP's image-text aligning property. Specifically, we used the whole class list of Pascal-VOC and computed the top-2 associated labels both for clean and perturbed images. We can observe the perturbations change the labels associated with the clean image.

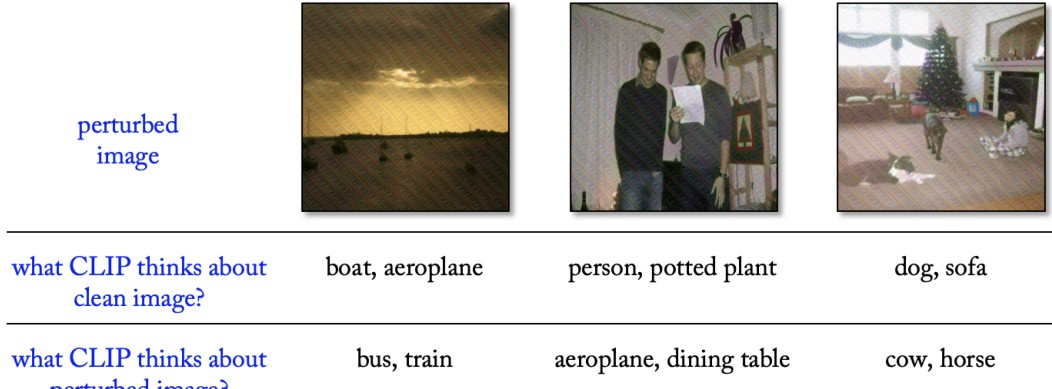

| | | | |
|---|---|---|---|
| perturbed image | | | |
| what CLIP thinks about clean image? | boat, aeroplane | person, potted plant | dog, sofa |
| what CLIP thinks about perturbed image? | bus, train | aeroplane, dining table | cow, horse |

Figure 2: Evaluation of adversarial images on CLIP. Surrogate model is VGG19 trained on Pascal-VOC.

**Mid-layer selection from surrogate model for training perturbation generator.** Our mid-layer from surrogate model is chosen based on the embedding size of CLIP: *e.g*. if the embedding size of the CLIP encoder is 512, we select the layer from the surrogate model that outputs 512 dimension features. In comparison, prior state-of-the-art generative attack TAP [1] *manually* searches for the optimal layer from the surrogate model to train the perturbation generator (see *Limitations* in Section 4.6 in their paper [1]). In particular, finding the optimal mid-layer (that gives the best attack results) requires searching over each block of $M$ layers (which is around an average of $M = 5$ layers [1]) for each surrogate model. Hence to find the best layer to train a perturbation generator for a particular model, the computation time cost for such an exhaustive search will be $MN$ GPU hours where $N$ is the total training time (in GPU hours) per layer. Moreover, our analysis shows this layer might not result in best attack results when the training data distribution varies and would require a manual search for all the different combinations of surrogate model and data distributions. Such a search is very time-consuming, impractical, and clearly not scalable. Finally, directly using TAP's suggested layer is not possible because the embedding size doesn't match that of CLIP, and would require us to introduce embedding modifications mechanisms (e.g. Principal Component Analysis (PCA), t-distributed stochastic neighbor embedding (t-SNE)) leading to an unreasonable increase in training time for every epoch. Note that if we do not consider the manual search of an optimal layer from the surrogate model to train the perturbation generator, then the proper baseline on ImageNet would be CDA [7]. As evident throughout our analysis, we convincingly outperform them on all settings.

## 3 Implementation Details

We use two multi-label datasets to stimulate the scenario of multi-object scenes: Pascal-VOC (training set: *trainval* from 'VOC2007' and 'VOC2012', testing set: 'VOC2007_test') and MS-COCO (training set: *train2017*, testing set: *val2017*). We follow prior works [7, 8] for the generator network for $\mathcal{G}_{\theta}(\cdot)$. To stabilize the training, we replace all the ReLU [16] activation functions with Fused Leaky ReLU [17] activation function (negative slope = 0.2, scale = $\sqrt{2}$). We use a margin $\alpha = 1.0$ for the contrastive loss. All our training setup uses ViT-B/16 as the CLIP model. We use Adam optimizer [18] with a learning rate $0.0001$, batch size 16, and exponential decay rates between 0.5 and 0.999. All images were resized to $224 \times 224$. Training time was observed to be $\sim$1 hr for Pascal-VOC dataset (10 epochs) and $\sim$10 hrs for MS-COCO dataset (5 epochs) on one NVIDIA GeForce RTX 3090 GPUs. PyTorch is employed [19] in all code implementations.