# OpenReview forum: "GAMA: Generative Adversarial Multi-Object Scene Attacks"
_NeurIPS.cc/2022/Conference — NeurIPS 2022 Accept_

### Official Review · Reviewer_SbKy · 2022-07-05

**Rating:** 6
**Confidence:** 3
**Soundness:** 3 good
**Presentation:** 3 good
**Contribution:** 3 good

**Summary:**

This paper proposes the first generative adversarial attack specifically for multi-object scene classification. It makes use of a CLIP model to provide additional losses during training. Good performance is achieved across a variety of tasks and models in black box and white box settings.

**Questions:**

1. Can you explain why you think the method works better when trained on Pascal-VOC vs MSCOCO?
2. Have you evaluated the adversarial images on the clip model?

**Limitations:**

The discussion of limitations and societal impacts is good.

**Strengths And Weaknesses:**

Strengths:
- The paper is well written and the explanation of the method is relatively easy to understand given its complexity.
- The results are convincing and demonstrate the effectiveness of the method on various tasks, datasets and models, and in both black and white box settings.
Weaknesses:
- The method does not outperform the baselines as consistently when trained on MSCOCO as it does on Pascal-VOC

---

> ### Author Response · Authors · 2022-08-02
> **Response to Comments by Reviewer SbKy**
>
> 1. **Response to Question 1:** We believe a possible explanation for this phenomenon is: Existing COCO image annotations have been shown to be extremely noisy (Section 3.4 of [A], also see [B]), and the multi-object classifiers are trained on these original (noisy) annotations. As a result, the image features computed by these classifiers have a mismatch with the image features computed by CLIP (that do not use these image annotations in any manner). Hence, this is probably creating a performance bottleneck.
>
>     [A] “TIDE: A General Toolbox for Identifying Object Detection Errors”, ECCV 2020
>
>     [B] “How I found nearly 300,000 errors in MS COCO” https://medium.com/@jamie_34747/how-i-found-nearly-300-000-errors-in-ms-coco-79d382edf22b
> ----
> 2. **Response to Question 2:** Based on the Reviewer’s suggestion, we evaluated the CLIP (as a “zero-shot prediction” model) on the perturbed images and computed the top two associated labels as suggested by CLIP in Figure 2 in Supplemental Material. We can observe that the perturbations change the labels associated with the clean image.

---

### Official Review · Reviewer_zurs · 2022-07-11

**Rating:** 7
**Confidence:** 5
**Soundness:** 3 good
**Presentation:** 3 good
**Contribution:** 3 good

**Summary:**

This paper studies the problem of adversarial attacks on multi-object scenes by leveraging a pre-trained vision language model. More specifically, it leverages the joint embedding space learned from cross-modal data to provide self-supervised signals for generating adversarial perturbations on the image domain. Experiments have been conducted on multiple image benchmarks including single-object benchmarks such as PASCAL-VOC and multi-object benchmarks such as MS-COCO.

**Questions:**

* [Q1] Explain why GAMA is less effective than TAP in some cases as mentioned in [W1].
* [Q2] Provide some findings and analysis of the GAMA perturbations (e.g., raw input level and feature level).

**Limitations:**

Limitations have been mentioned in the paper.

**Strengths And Weaknesses:**

Strengths:
* [S1] This paper presents a clean solution to generating multi-object attacks. The quantitative results demonstrate the strength of the proposed attacks when compared to state-of-the-art attacks. The reviewer feels that the paper could possibly open up a new research area of using foundation models to craft adversarial attacks and defenses in the future.

Weaknesses:
* [W1] In Table 3 and Table 5, the proposed GAMA is less effective than TAP [Ref 12] when applied to the VGG19 network. It would be helpful to provide some explanations and high-level insights on this.
* [W2] Although the quantitative experiments are thorough and solid enough, the qualitative analysis is definitely insufficient (Figure 5). As a scientific study, it would be helpful to understand how GAMA explores the perturbation space compared to other existing attacks (e.g., perturbation/feature visualization).
* [W3] Missing discussions on mid-level or semantic-level adversarial attacks using generative models ([NewRef1-2]).

References
* [NewRef 1] SemanticAdv: Generating Adversarial Examples via Attribute-conditioned Image Editing. Qiu et al., In ECCV 2020.
* [NewRef 2] Unrestricted Adversarial Examples via Semantic Manipulation. Bhattad et al., In ICLR 2020.

---

> ### Author Response · Authors · 2022-08-02
> **Response to Comments by Reviewer zurs**
>
> 1. **Response to W1/Q1:**
>     - We attribute this performance gap to the optimal mid-level layer (from the surrogate model, chosen to compute the learning loss) that is being searched manually for TAP as pointed out by the authors of TAP (see Limitations in their paper). This manual search (*a*) is extremely time-consuming and not scalable as it has to be done for every combination of surrogate model and data distribution and (*b*) is data distribution specific as manual layers (chosen for ImageNet in [12]) do not yield the same level of attack performance when trained with different data distributions (see L265-L270). Our mid-layer is chosen based on the embedding size of CLIP: e.g. if the embedding size of the CLIP encoder is 512, we select the layer from the surrogate model that outputs 512 dimension features.
>      - The best attack method that doesn’t need such a manual layer search is CDA [11]. But we convincingly outperform them in all settings as shown in Table 2-7.
> ----
> 2. **Response to W2/Q2:** Thank you for the suggestion. Due to the limitation of space in the main paper, we provided the feature visualization comparison with TAP [12] for both MS-COCO and Pascal VOC for cross-domain black-box attacks in Figure 1 of Supplementary Material.
> ----
> 3. **Response to W3:** Thank you for pointing these out. We have added and discussed them in the revised version (L88 and L105-106, New Ref 1 is [61], NewRef 2 is [60]).

---

> > ### Comment · Reviewer_zurs · 2022-08-07
> > **Thanks for the responses!**
> >
> > It would be great to include the exhaustive search of the mid-level layer of TAP in the discussion. Ideally, having a metric to quantify the computational complexity of these methods will clarify things much better.

---

> > > ### Author Response · Authors · 2022-08-08
> > > **Post-rebuttal response**
> > >
> > > Thank you for taking the time to read our rebuttal. Please feel free to further raise your concerns in case you have any.
> > >
> > > ----
> > > - "***It would be great to include the exhaustive search of the mid-level layer of TAP in the discussion.***"
> > >
> > >     Thank you for the suggestion. We have included a discussion on exhaustive search of the mid-level layer of TAP in the Supplementary Material (L59-L76).
> > > ----
> > > - "***Ideally, having a metric to quantify the computational complexity of these methods will clarify things much better.***"
> > >
> > >     The search for the optimal mid-layer (that gives the best attack results) requires searching over each block of *K* layers (which is around an average of 5 layers [12]) for each surrogate model. Hence to find the best layer to train a perturbation generator for a particular model as per [12], the computation time cost will be *KN* GPU hours (where *N* is the total training time (in GPU hours) for one layer) for **every** training data distribution and for **every** surrogate model. Furthermore, our analysis shows that the resultant optimal layer might not be the best one when the training data distribution varies (as shown in the analysis in our paper). On the contrary, our method doesn’t depend on such exhaustive search for best attack results.

---

### Official Review · Reviewer_pwfE · 2022-07-17

**Rating:** 6
**Confidence:** 4
**Soundness:** 3 good
**Presentation:** 4 excellent
**Contribution:** 3 good

**Summary:**

This paper introduces the first multi-object generative attack, GAMA, which utilizes a pre-trained CLIP model as an attacker tool to enhance the transferability across different data distributions. Extensive experiments in this paper show that GAMA can achieve state-of-the-art transferability on various black-box settings when training on multi-label datasets. Besides, GAMA also shows its superior efficacy against various defense methods compared with baselines.

**Questions:**

1. See Con 1), How does GAMA perform and transfer compared with prior works on attacking single-object scenes? This result should play an important role in judging the significance and applicability of this paper to the community. If GAMA can still beat those baselines that focus on single-object attacks then in the future one could directly choose to use GAMA to perform attacks without caring about the number of objects in the image.

2. For $\rho\_\text{img}$ and $\rho\_\text{txt}$, are you using the joint image/text embeddings or the embeddings that directly come from the image/text encoder? Do you use any normalization on $z$, $\hat{z}$, $\rho\_\text{img}$, and $\rho\_\text{txt}$? If not, how do you ensure that $\hat{z}$, $\rho\_\text{img}$, and $\rho\_\text{txt}$ are within the same feature space so that you can contrast them with each other, considering that the surrogate classifier and CLIP image/text encoder are using different architectures.

**Limitations:**

The authors adequately discussed the limitations and potential societal impact in the last section of the main paper.

**Strengths And Weaknesses:**

Pros:

1. This paper first proposes the problem of multi-object scene based adversarial attack.

2. Introducing joint vision-and-language pre-trained models, such as CLIP, to adversarial attacks is interesting. Specifically, the authors generate multi-class text prompts and leverage the semantics relationship underlying the text representations to tackle the multi-object attack problem.

3. When training on multi-label datasets, the proposed method outperforms prior works in terms of transferability on various black-box settings.

4. This paper is clearly written and very easy to follow.

Cons:

1. This paper only considers the settings of training on multi-object datasets in the experiments, such as Pascal-VOC and MS-COCO, and shows its superiority compared with prior works. Those prior works focus on single-object scenes and the authors adapted them to multi-object scenes. However, the proposed method should also be able to handle single-object scenarios, while the experiments of training on single-object scenes, such as ImageNet, and transferring to other single-object datasets, as done in TAP [12], BIA [13], are not included in this paper. I am curious about how GAMA performs compared with those prior works on this standard setting.

2. Typos: L159 and L201: $\mathcal{L}\_{\text{txt}}$ -> $\mathcal{L}\_{\text{img}}$.

---

> ### Author Response · Authors · 2022-08-02
> **Response to Comments by Reviewer pwfE**
>
> 2. **Comparison when trained with a single-object dataset:** Based on the Reviewer’s suggestion:
>     - We analyze the average performance on the single-object dataset ImageNet for GAMA and compare it with two SOTA generative attacks: CDA [11] - no manual optimal layer search from the surrogate model is required for training the perturbation generator, and TAP [12] - manual optimal layer search required for best attack performance.
>     - We train on ImageNet with DenseNet169 as a surrogate model. We then evaluate the attacks for six victim models for datasets ImageNet/Pascal-VOC/MS-COCO and one victim model for datasets CIFAR10/CIFAR100.  Lower is better.
>
> |  | ImageNet | Pascal-VOC | MS-COCO | CIFAR10 | CIFAR100 |
> | --- | --- | --- | --- | --- | --- |
> | CDA[11] | 33.65 | 36.22 | 26.98 | 85.01 | 54.71
> | TAP[12] | **09.46** | 24.84 | 17.51 | 82.35 | 49.38
> | Ours | 21.19 | **20.89** | **14.50** | **75.49** | **43.60**
>
>    Our results are better than CDA on all datasets and better than TAP on all datasets except Imagenet, which is explained below.
>
> - Other than ImageNet, our results on all the datasets are better than TAP (average TAP/Ours- 50.73%/38.62%). Our results are poorer than TAP on ImageNet because TAP manually searches for the optimal layer from the surrogate model to train the generator (see *Limitation* in [12]). Such a search is very time-consuming, impractical, and clearly not scalable. Our method doesn’t rely on manually finding such an optimal layer as our mid-layer is decided by looking at the embedding size of CLIP features. Next, directly using TAP’s suggested layer is not possible as the embedding size doesn’t match that of CLIP, and would require us to introduce embedding modifications (e.g. PCA/tSNE) leading to an unreasonable increase in training time. Finally, TAP shows degradation in performance when distribution changes from ImageNet (as shown in our paper) and would still require a manual search for all the different combinations of surrogate model and data distributions we have explored in this work.
>
> - If we do not consider the manual search of an optimal layer from the surrogate model to train the generator, then the proper baseline on ImageNet would be CDA [11]. The average attack performance (over six models) for ImageNet is: CDA/Ours = 33.65%/21.19%, and we convincingly outperform them on all other settings.
> ----
> 2. **Typos:** Thank you for pointing these out. We have corrected them in the revised version.
> ----
> 3. **Normalization of embeddings:** We are using the embeddings directly from the image/text encoder. Yes, they are normalized before using them in the proposed loss functions.  We have highlighted this in the revised version (L213).

---

> > ### Comment · Reviewer_pwfE · 2022-08-09
> > **Thanks for your response**
> >
> > Thanks for the response from the authors. It is great to see the method performs well on single-object datasets. My concerns are addressed.

---

> > > ### Author Response · Authors · 2022-08-09
> > > **Post-rebuttal response**
> > >
> > > Thank you for taking the time to read our response. We are happy to see your concerns are addressed.
> > >
> > > Best Wishes,
> > >
> > > Authors

---

### Official Review · Reviewer_FDtQ · 2022-07-18

**Rating:** 5
**Confidence:** 4
**Soundness:** 3 good
**Presentation:** 3 good
**Contribution:** 2 fair

**Summary:**

This paper proposes the GAMA attack, a generative approach to generating adversarial examples. The proposed method incorporates the vision-language model CLIP in the training of the generator. Experiments demonstrate the effectiveness of the proposed method in various attack settings.

**Questions:**

* Does the GAMA attack perform well on the single-object dataset like ImageNet? In Table 1, it is said that the GAMA attack can analyze both attacking scenarios with input scenes that contain multiple objects or a single object. However, in experiments, the GAMA attack is only trained on PASCAL-VOC and MS-COCO, which are scenes containing multiple objects. So does the GAMA attack perform well when training on the single-object dataset like ImageNet?
* How to conduct fair comparisons with baseline methods, which aim to handle attacking scenarios with input scenes that contain a single object? As mentioned in Table 1, previous methods only analyze single-object attacking scenarios. However, in experiments, all previous methods are also trained on PASCAL-VOC and MS-COCO,  which are scenes containing multiple objects. Is it fair to compare with baseline methods in a different setting from their original design?

**Limitations:**

The authors adequately addressed the limitations and potential negative social impact of their work.

**Strengths And Weaknesses:**

### Strengths
* The paper is well-written and easy to follow.
* Experiments demonstrate the effectiveness of the proposed method in various attack settings, including both the white-box and the black-box, even under the setting with different datasets and tasks.

### Weaknesses
* The major concern is that the algorithmic contribution of the proposed method is limited. The main contribution of the proposed method is incorporating the vision-language model CLIP in the training of the generator, which is also the main concern.  Previous methods only train the generator with a pre-trained DNN and the corresponding dataset. Instead of a pre-trained model, the proposed method requires access to the CLIP model. The  CLIP  extracts knowledge from ~400 million image-text pairs. Thus, the proposed method actually leverages much more information in training the generator, compared with previous generative approaches. Thus, the effectiveness of the proposed method is obvious, since it leverages much more information. From this perspective, the algorithmic contribution of the proposed method seems limited.

---

> ### Author Response · Authors · 2022-08-02
> **Response to Comments by Reviewer FDtQ (part 1)**
>
> 1. **Algorithmic contribution:** The overwhelming majority of work in generative adversarial attacks [10-13] has been on single-object images. Recently, there have been methods [26-30] for attacking multi-object images which are more natural or better representatives of real-world scenes. However, all these works are image-specific approaches. Our work proposes the first method for attacks on multi-object images in a generative setup, which is the main contribution of the paper. Our approach for crafting perturbations builds on pre-trained DNNs but incorporates open-source vision-language models like CLIP to encode the relationships between multiple objects in the scene *via* language derivatives. The effectiveness of our approach does not arise because we use CLIP. Rather, we use CLIP because we work on far more complex images than existing approaches (multi-object vs single-object images) and we need a tool like CLIP to understand the relationships between the objects.
>     More specifically, the use of CLIP in our method is non-trivial.
>
>     - In order to understand the “contextual information” relationships between the different objects in the multi-object scenes, we leverage CLIP’s inherent ability to encode text into features for representing this “context information” in the multi-object images through language derivatives.
>     - As CLIP is trained on ~400 million “image-text pairs”, it aligns the context encoded in language space with context from image space. Our novelty is in exploiting this aligning property to our advantage and **misaligning** the perturbed image w.r.t. the context captured in text features.
> ----
> 2. **Comparison when trained with a single-object dataset:** Regarding Table 1, we mean that it can handle a data distribution (Pascal-VOC and MS-COCO) containing both multiple object images and single object images. Based on the Reviewer’s suggestion:
>     - We analyze the average performance on the single-object dataset ImageNet for GAMA and compare it with two SOTA generative attacks: CDA [11] - no manual optimal layer search from the surrogate model is required for training the perturbation generator, and TAP [12] - manual optimal layer search required for best attack performance.
>     - We train on ImageNet with DenseNet169 as a surrogate model. We then evaluate the attacks for six victim models for datasets ImageNet/Pascal-VOC/MS-COCO and one victim model for datasets CIFAR10/CIFAR100.  Lower is better.
>
> |  | ImageNet | Pascal-VOC | MS-COCO | CIFAR10 | CIFAR100 |
> | --- | --- | --- | --- | --- | --- |
> | CDA[11] | 33.65 | 36.22 | 26.98 | 85.01 | 54.71
> | TAP[12] | **09.46** | 24.84 | 17.51 | 82.35 | 49.38
> | Ours | 21.19 | **20.89** | **14.50** | **75.49** | **43.60**
>
>    Our results are better than CDA on all datasets and better than TAP on all datasets except Imagenet, which is explained below.
>
> - Other than ImageNet, our results on all the datasets are better than TAP (average TAP/Ours- 50.73%/38.62%). Our results are poorer than TAP on ImageNet because TAP manually searches for the optimal layer from the surrogate model to train the generator (see *Limitation* in [12]). Such a search is very time-consuming, impractical, and clearly not scalable. Our method doesn’t rely on manually finding such an optimal layer as our mid-layer is decided by looking at the embedding size of CLIP features. Next, directly using TAP’s suggested layer is not possible as the embedding size doesn’t match that of CLIP, and would require us to introduce embedding modifications (e.g. PCA/tSNE) leading to an unreasonable increase in training time. Finally, TAP shows degradation in performance when distribution changes from ImageNet (as shown in our paper) and would still require a manual search for all the different combinations of surrogate model and data distributions we have explored in this work.
>
> - If we do not consider the manual search of an optimal layer from the surrogate model to train the generator, then the proper baseline on ImageNet would be CDA [11]. The average attack performance (over six models) for ImageNet is: CDA/Ours = 33.65%/21.19%, and we convincingly outperform them on other all settings.
>
> ----

---

> > ### Author Response · Authors · 2022-08-02
> > **Response to Comments by Reviewer FDtQ (part 2)**
> >
> > 3. **Fairness of baselines:** Since our method is the first that considers generative approaches for crafting attacks on multi-object scenes, we need to adapt existing methods to be able to compare fairly. We have two options.
> >     - *Option 1*: First, we could use existing methods trained on single-object images and use them to craft attacks on multi-object images. This is obviously unfair since these methods were never aware of the characteristics of multi-object images.
> >     - *Option 2*: Second, we can make suitable adjustments to the prior attack algorithms (see L233-236) to train with multi-object scenes and ask them to create perturbations on multi-object scenes. This is a fairer approach than *Option 1* as the attacks now use multi-object images during training their perturbation generator. Results of this approach are shown in Tables 2-7 in the main paper.
> >
> >     Hence, it is fair to compare prior generative attacks under *Option 2* as done in our paper.

---

### Author Response · Authors · 2022-08-02
**Official Response**

We are very thankful to the reviewers for their thorough reviews and encouraging comments.  We have uploaded the revised version of the main manuscript and supplementary material with the following changes (highlighted in blue):

1. Typos pointed out by Reviewer *pwfE* are corrected.
2. L213: Statement added to indicated embeddings are normalized before loss functions are computed as suggested by Reviewer *pwfE*.
3. NewRef 1 and NewRef 2 suggested by Reviewer *zurs* are added and discussed in the Related Works section.
4. L53-L57: Added paragraph and Figure 2 in Supplementary Material for response to Reviewer *SbKy*.

We can further incorporate other changes that the reviewers might suggest based on the post-rebuttal discussion. We now provide detailed responses to each reviewer's queries. We look forward to addressing their subsequent comments.

----

*Post-rebuttal edits to manuscript*:
1. A discussion on mid-layer selection from surrogate model (in comparison to prior works) has been added to Supplementary Material (L59-76) as suggested by Reviewer *zurs*.

---

### Author Response · Authors · 2022-08-07
**Discussion on further concerns**

Hello Reviewers,

Thank you again for your helpful and insightful comments. We would be happy to address any further concerns you have based on our rebuttal responses.


Best wishes,

Authors

---

### Meta-Review · Area_Chair_R23b · 2022-08-23

**Recommendation:** Accept
**Confidence:** Certain

**Metareview:**

The authors proposed the first multi-object generative attack, GAMA, which utilizes the vision-language model CLIP as an attacker's tool in the training of the generator to enhance the transferability across different data distributions.
All four reviewers recognize that this paper is well-written and easy to follow. The presented results also are promising. Most importantly, the Generative Adversarial Multi-object scene Attack is good direction for further study.
Since the four reviewers consistently accept the paper with good comments, the AC made a decision of acceptance.

**Award:**

No

---

### Decision · Program_Chairs · 2022-09-14

Accept